# The Quest for the Nature of the Dark Matter: The Need of a New Paradigm

**Fabrizio Nesti** [1,2], **Paolo Salucci** [3,4,*] and **Nicola Turini** [5,6]

1 Dipartimento di Scienze Fisiche e Chimiche, Università dell'Aquila, Via Vetoio, 67100 L'Aquila, Italy; fabrizio.nesti@aquila.infn.it
2 INFN, Laboratori Nazionali del Gran Sasso, Assergi, 67100 L'Aquila, Italy
3 SISSA-ISAS—International School for Advanced Studies, Via Bonomea 265, 34136 Trieste, Italy
4 INFN, Trieste, Iniziativa Specifica QGSKY, Via Valerio, 34127 Trieste, Italy
5 Physics Department, Università di Siena, Via Roma 56, 53100 Siena, Italy; nicola.turini@unisi.it
6 CERN, CH-1211 Geneva, Switzerland
* Correspondence: salucci@sissa.com

**Abstract:** The phenomenon of the Dark matter baffles the researchers: the underlying dark particle has escaped so far the detection and its astrophysical role appears complex and entangled with that of the standard luminous particles. We propose that, in order to act efficiently, alongside with abandoning the current $\Lambda CDM$ scenario, we need also to shift the Paradigm from which it emerged.

**Keywords:** dark matter; galaxy structure; cosmology

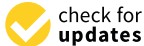



## 1. The Phenomenon of Dark Matter

The phenomenon of the Dark Matter is one of the most intriguing mysteries in the Universe. In fact, not only it implies the existence of unknown Science and in particular of unknown Physics, but it concerns the fabrics itself of the Universe. A new Law of Nature, yet to be discovered, seems to be at work. As Zwicky found back in the 30's [1] and Vera Rubin in the late 70's [2], the law of Gravity seems to fail in Clusters of Galaxies and in (Disk) Galaxies. Especially in the latter, one detects large anomalous motions: the stars and the gaseus component in a galaxy do not move as they should do under their own gravity, but as they were attracted by something of invisible.

Disk systems can be divided in normal spirals, dwarf irregulars and Low Surface Brightness galaxies. In all these objects, the equilibrium between the force of Gravity and the motions that oppose it, has a simple realization: the stars (and the subdominant HI gas) rotate around the galaxy center. However, we realize[1] that such rotation is very much unrelated to the spatial distribution of the stars and gas, in strong disagreement with the Newton Law of Gravity.

The objects belonging to the above most common types of galaxies are relatively simple to investigate, in that we have:

$$R \frac{d\Phi(R)}{dR} = V^2(R) \tag{1}$$

where the (measured) circular velocity and the total galaxy gravitational potential are indicated by: $V(R)$ and $\Phi(R)$. A disk of stars is their main luminous component, whose surface mass density $\Sigma_\star(R)$,[2] proportional to the surface luminosity measured by the photometry, is well approximated by [3]:

$$\Sigma_\star(r) = \frac{M_D}{2\pi R_D^2} e^{-R/R_D} \tag{2}$$

where $M_D$ is the mass of the stellar disk to be determined and $R_D$ is its scale length measured from the photometry. At $R \geq 3\,R_D$, in all objects in similar fashion, this component rapidly disappears, so that $R_D$ plays the role of the characteristic radius of the stellar matter. Equation (1) together with the Poisson equation for this component (in cylindrical coordinates, $\delta$ is the Kronecker function):

$$\Delta\Phi_\star(R,z) = 4\pi G\,\Sigma_\star(R)\delta(z) \tag{3}$$

yields $V_\star(y)$, the luminous matter contribution to the circular velocity $V(R)$. With $y \equiv R/R_D$ and

$$v_\star^2(y) \equiv \frac{G^{-1}V_\star^2(y)R_D}{M_D}$$

we have ($I$, $K$ are the Bessel functions evaluated at $y/2$):

$$v_\star^2(y) = \frac{1}{2}\,y^2(I_0\,K_0 - I_1\,K_1)|_{y/2} \tag{4}$$

It is interesting to briefly show the dynamical evidence for the presence of a DM halo in the above rotating galaxies and how to derive its spatial density. Defining $\nabla \equiv d\log V / d\log R$, from the above equations, we have: $\nabla_\star(y) \simeq 0.87 - 0.5\,y + 0.043\,y^2$. According to Newtonian gravity one should expect: $\nabla(y) = \nabla_\star(y)$, instead, we find: $\nabla(y) > \nabla_\star(y)$ (a) at all radii $y$ in galaxies with steep rotation curves ($\nabla(2) > 0.5$) and (b) for $y > 2$, in galaxies with a flatter RC (e.g. see Figure 1). In order to restore the law of Gravity one adds a "spherical dark halo" component for which:[3]

$$V_h^2(y) = -V_\star^2(y) + V^2(y) \tag{5}$$

with the constraint:

$$\nabla_h(y) = \frac{\nabla(y)\,V^2(y) - \nabla_\star(y)\,V_\star^2(y)}{V^2(y) - V_\star^2(y)} \tag{6}$$

Then, we have: $V_h^2(R) = G\frac{\int 4\pi\rho_h(R)R^2 dR}{R}$ with $\rho_h(R)$ the DM halo density.

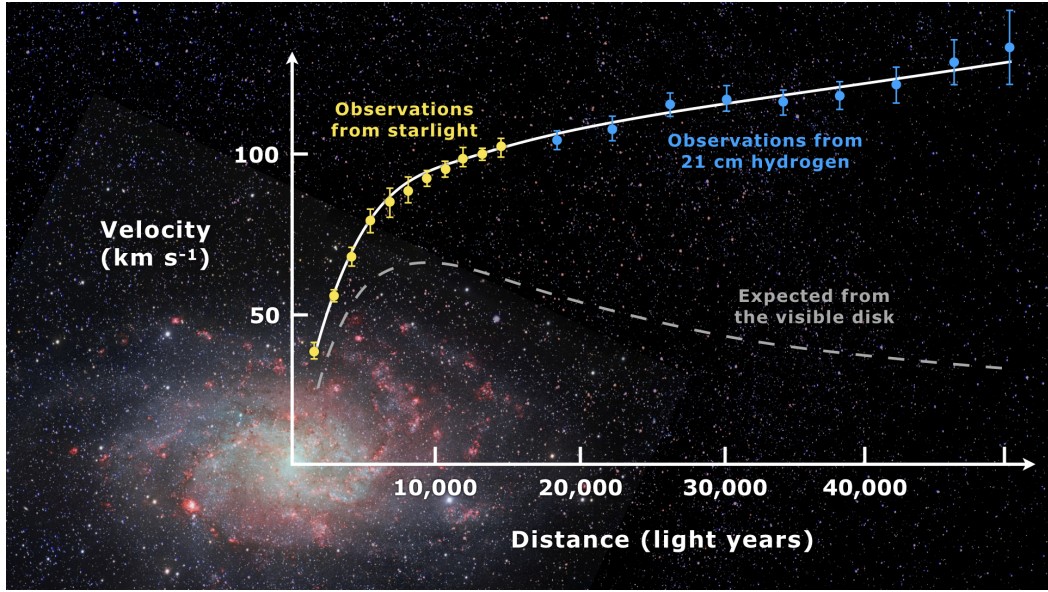

**Figure 1.** M33: the profile of the stellar disk contribution to the circular velocity does not coincide with the profile of the latter, being at all radii: $\nabla > \nabla_\star$ (from [4]).

It is well known that the above aspects of the "Dark Matter Phenomenon" are present also in the other types of galaxies (see, e.g., [5]) and imply the existence of a massive particle

that does not interact with the standard matter via electromagnetic or strong force. Remarkably, these are also needed to explain a number of cosmological observations including the rate of the expansion of the universe, the anisotropies in the Cosmic Background Radiation, the properties of the large scale structures, the phenomenology of the gravitational lensing of distant galaxies by nearby clusters of galaxies, and the existence itself of galaxies (e.g., see: [6,7]) [4].

The starting point to account for all the above has been, therefore, to postulate the ubiquitous presence in the Universe of massive particles that emit radiation at a level which is totally negligible with respect to that emitted by the Standard Model (SM) particles. Then, this particle, by definition beyond the SM, is hidden to us also when it aggregates in vast amounts. As matter of fact, we take the dark particle option as a foundation of Physics and Cosmology. However, one has to stress that this does not automatically lead to infer neither the mass nor the nature itself of such a particle. Furthermore, the present status of "darkness" means that the particle has a very small, but not necessarily zero, self-interactions or interactions with the SM particles, with a number of cosmological, physical and astrophysical consequences.

**2. The Standard Paradigm for the Dark Matter Phenomenon**

The next step of the investigation has been to endow the particle behind the Dark Matter Phenomenon with a theoretical scenario. First, let us introduce the concept of a "Paradigm for the Dark Matter Phenomenon", i.e., a refined set of properties that the *actual* DM scenario must possess and that, in turn, reveals the nature itself of the dark particle. After the first "detections" of DM in the Universe, a Paradigm has, indeed, emerged and lasted until today. According to this paradigm, the correct *scenario* behind the DM Phenomenon must have the following properties:

1.  it connects the (new) Dark Matter physics with the (known) physics of the Early Universe; it introduces in a natural way the required massive dark particle and relates it with the value of the cosmological mass density of the expanding Universe.
2.  it is mathematically described by a very small number of parameters and by a very well known and specific initial conditions, while having, at the same time, a strong predictive power on the evolution of the structures of the Universe. Furthermore (and far than being obvious), such evolution can be thoroughly followed by proper numerical simulations.
3.  its (unique) dark particle can be detected by experiments and observations with the present technology.
4.  it sheds light on issues of the Standard Model particle physics.
5.  it provides us with hints for solving long standing big issues of Physics.

In other words, the ruling paradigm heads us towards scenarios for the dark matter phenomenon that are very beautiful, hopefully towards the most beautiful one, where beauty is meant in the sense of simplicity, naturalness, usefulness, achieving expectations and harmonically extending our knowledge. For definiteness and clarity of the discussion, we name this paradigm as: "The Apollonian paradigm". Let us point out that, in doing so, we just *name* concepts emerged and solidified in the mid 80' and that, since then, were used as lighthouses in the investigation of the DM mystery. Continuing our narration, this procedure has resulted very successful: the above Paradigm has straightforwardly led Cosmologists to one specific scenario, the well known ΛCDM (e.g., [6,7]), that proved able to reproduce several crucial aspects of the DMP.

Let us also stress that the Apollonian paradigm, as a consequence of its definition, in addition to providing us with a very strong candidate for the actual scenario behind the dark particle, is also linked very directly to the (in)successes that the latter has in reproducing the DMP. Thus, to adopt a-priori the above scenario or to adhere to the originating paradigm, it is conceptually the same thing. Finally, the ΛCDM scenario is rather unique: in the past 30 years no other scenario has emerged with such complete Apollonian status.

$\Lambda$ stays for the Dark Energy having 70% of the total energy density of the Universe and CDM for Cold Dark Matter. Cold refers to the fact that the dark matter particles move very slowly compared to the speed of light. Dark means that these particles, in normal circumstances, do not interact with the ordinary matter via electromagnetic force but very feebly with a cross section of the order of $3 \times 10^{-26}$ cm$^3$/s characteristic of the Weak Force. This specific value of the cross section inserted in the Physics of the early Universe, makes the predicted WIMP (Weak Interacting Massive Particles) relic density compatible with the observed value of about $3 \times 10^{-30}$ g/cm$^3$ (e.g., [6]).

Among the CDM particles, all in line with the above paradigm, we must stress the prominent role is taken by the one that the (much favoured) Supersymmetry theory has inside his corpus: i.e., the Neutralino. To choose this particle brings also the bonus of explaining, in one shot, the existence of the DM particle, its relic density, and the presumed "naturalness problem" of the Standard Model. It is well known that the recognised beauty of this theory has been the main motivation for searching the related particle by means of numerous observational and experimental programs of Fundamental Physics.

In this scenario, the density perturbations evolve through a series of halos mergings from the smallest to the biggest in mass, the final state being a matrioska of halos with smaller halos inside bigger ones. Very distinctively, these dark halos show an universal spherical spatial density [8]:

$$\rho_{NFW}(r) = \frac{\rho_s}{(r/r_s)(1 + r/r_s)^2} \tag{7}$$

where $r_s$ is a characteristic inner radius, and $\rho_s$ the related density. Notably, this scenario confirms its beauty, and turns out to be extremely falsifiable since in all the Universe and throughout its history, the related dark component has to create structures with the same configuration.

Now, the well known situation is that no such dark particle has been detected in the past 30 years. This has occurred in experiments at underground laboratories, searching for the soft scatter of these particles with particular nucleus; in particle collisions at LHC collider with a general search for Supersymmetric partners or more exotic invisible particles to be seen as missing momentum of unbalanced collision events; in measurements at space observatories as gamma rays coming from dense regions of the Universe where the dark particle should annihilate with its antiparticle (see e.g., [9,10]). Furthermore, the current upper limits for the energy scale of SuSy, as indicated by LHC experiments, rules out the Neutralino as the DM particle. Nevertheless, a WIMP particle from Effective Field Theories, outside the SuSy environment, can be still proposed for detection experiments.

It is, important to notice that, in the attempts made so far, only WIMP particles have been thoroughly searched. The search for particles related to other DM scenarios has been very limited and almost no blind search has been performed. Thus, the lack of detection of the dark particle so far, in no way indicates that this does not exist, but just indicates the failure of the detection strategies related to particular scenarios.

In recent years, at different cosmological scales, observational evidence in strong tension with the above scenario has emerged (e.g., [11,12]). Here, we focus on the distribution of dark matter in galaxies, a topic for which the failure of the $\Lambda$CDM scenario is the most eventful and striking [13]. Dark Matter is, in fact, located mostly in galaxies that come with very large ranges of total masses, luminosities, sizes, dynamical state and morphologies. While each of them is a laboratory for the new physics, the diversity of the properties of their luminous components is an asset for the investigation of their dark components.

## 3. The Cored DM Halos

The rotation curves of disk systems are well measured from the Doppler measurements of the H$_\alpha$ and the 21 cm galaxy emission lines. In many cases they extend well beyond the stellar disk edge and, in some case, out to 20% of the dark halo size. In the outermost regions of the dark halos, devoid of rotating stars and HI gas, we have other useful tracers

of the galaxy mass profile; the latter is available, from the galaxy center to the edge of the dark matter halo, for a sufficient number of disk systems [14]. By investigating several thousands RCs covering: (a) all the *morphologies* of the disk systems: normal spirals, dwarf irregulars and low surface brightness galaxies and (b) all the values of their *magnitudes* from the faintest to the most luminous objects, one finds that the RCs, combine in an Universal Rotation Curve (see [5]) defined from the center of the galaxies out to the edge of the dark matter halos.

Specifically, for the disk systems of the local Universe, in the pipeline set to retrieve the galaxy dark and luminous mass distributions from their circular velocities $V(R)$, the great majority of the latter can be represented by an unique function $V_{URC}(R/R_D, Mag, C, T)$, where, for each object, $R_D$ is the disk length scale of Equation (2), $Mag$ is the magnitude, $C$ indicates how compact its distribution of light is and T the Hubble morphology (see Figures 4 and 5 in the pioneering work by Rubin and collaborators [15] and the subsequent URC series of papers [14,16–21]). Individual features in the RCs are sometime present, but they originate from physical phenomenons (such as non circular motions, non exponential stellar disks, presence of (small) bulges and bars etc.) that are not directly related to the DM phenomenon and get mostly damped out by the stacking procedure.

$V_{coadd}(R/R_D, P_i)$ and $\delta V_{coadd}(R/R_D, P_i)$, the coadded velocity data and their r.m.s. (the points with errorbars in Figure 2) are obtained by stacking, with a proper procedure, a large number of individual RCs in bins of the observed quantity(ies) $P_i$, (e.g., $Mag$ and $T$). $V_{URC}(R/R_D, P_i)$, the ensemble of solid lines in Figure 2, is an analytical function found to fit the $V_{coadd}$ data (see [18]). Let us stress that the $V_{coadd}$ and their r.m.s. $\delta V_{coadd}$ are crucial kinematical quantities; first, since $\delta V_{coadd} \ll V_{coadd}$, the latter provide us with excellent *templates* for a very large majority of the *individual* RCs of the disk systems. Furthermore, the analytical function $V_{URC}$, derived from the $V_{coadd}$, allows one to interpret this set of data in terms of a universal mass model.

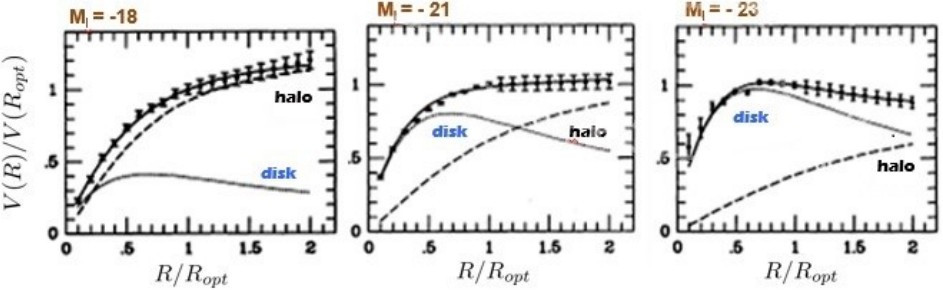

**Figure 2.** Stacking of 1000 individual RCs in 3 typical luminosity bins. The coadded curves $V(R/R_{opt})/V(R_{opt})$ (points with error bars) are fitted with the URC model (solid line) which includes a cored DM halo (dashed line) and a Freeman Disk (dotted line) (see [18,22]).

Remarkably, all the identifying quantities of the RCs (i.e., $Mag, T, R_D$) belong to the *stellar* component of the galaxies despite that the *dark* component dominates the mass distribution. This is a first indication of a *direct coupling* between the dark and luminous components.

The proposed mass model features the following two components: the above stellar disk with mass $M_D$ as a free parameter and a dark halo with the Burkert density profile [22]:

$$\rho_B(r) = \frac{\rho_0}{(1 + r/r_0)(1 + (r/r_0)^2)} \tag{8}$$

This profile has 2 free parameters like the NFW profile (but with different physical meanings): the central density $\rho_0$ and the core radius $r_0$ that marks the edge of the region in which the DM density is roughly constant.

The stellar disk + Burkert halo model reproduces well the coadded RCs [17–24] and also individual RCs of disk galaxies (see also [5]) and it is dubbed as the URC model.

Notably, for the Burkert and any other halo density profile with a core of size $a$, we have:

$$\nabla_h = \kappa \, a / R_D \tag{9}$$

with $\kappa$ a constant depending of the density profile. Its success highlights the failure of the NFW halo + stellar disk mass model in reproducing the *coadded* RCs [25], so as (almost) the totality of the available high quality *individual* RCs (e.g., [26–35]). Such a failure is very serious in that one often finds, for the NFW halo + Freeman disk mass model, not only bad fits, but also implausible best-fitting values for the masses of the stellar disk and of the dark halo and for the two structural parameters of the NFW halo (see, e.g., [25]).

This raises strong doubts about the collisionless status itself of the DM particles in galaxies, a fundamental aspect of the $\Lambda CDM$ scenario. Furthermore, at radii $r \gg r_0$, the density profile of the dark matter halos of disk galaxies falls back to be that of the collisionless particles [14] (see Figure 3). These facts fit well with the above observational scenario: in the outermost regions of halos, the luminous and dark matter are so rarefied that, in the past 10 Gyrs, had no time to interact appreciably among themselves, even if this were physically allowed. Thus, on the scale of the halo's virial radius, the standard physics of galaxy formation is not in tension with the observed distribution of dark matter. Differently, on the scales of the distribution of the luminous component, observation imply that the DM halo density has undergone a significant and not yet well understood evolution over the Hubble time (see also [36]).

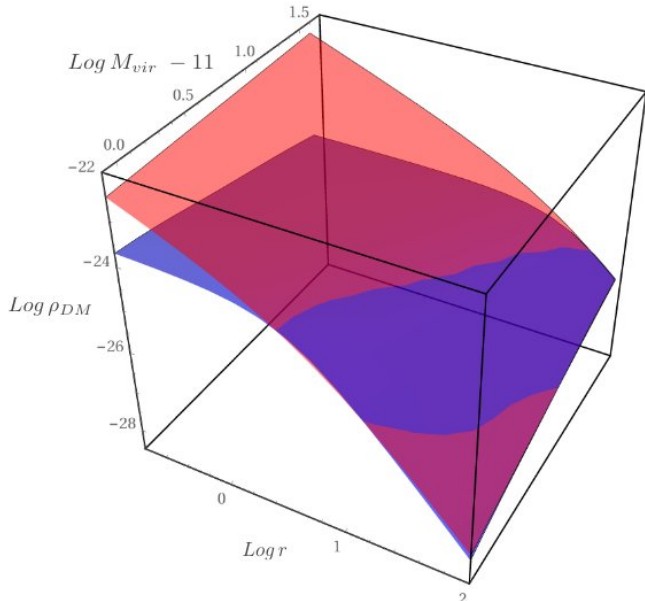

**Figure 3.** The density of the DM halos today (blue) and the (extrapolated) primordial one (red) as function of radius and halo mass (from [14]). The agreement of the two density profiles at outer radii reveals a time evolution of the density of the central regions of the DM halo. Log-units: kpc g/cm$^3$, $10^{11}$ M$_\odot$.

The mass distribution of a disk galaxy is described, in principle, by three parameters: one belonging to the luminous world and two to the dark one, representing structural quantities not existing in the standard $\Lambda$CDM scenario.

In disk galaxies a further extraordinary observational evidence emerges: the three parameters $r_0$ $\rho_0$ and $M_D$ result well correlated among themselves (see Figure 4, [14] and Figure 11 in [5]), which poses the basis of the URC model. It is important to realize that the above correlations cannot occur in the standard $\Lambda$CDM scenario, and should then thus a crucial subject of investigation. In next section, therefore, we focus on these evidences and on the resulting structural physical properties of disk galaxies.

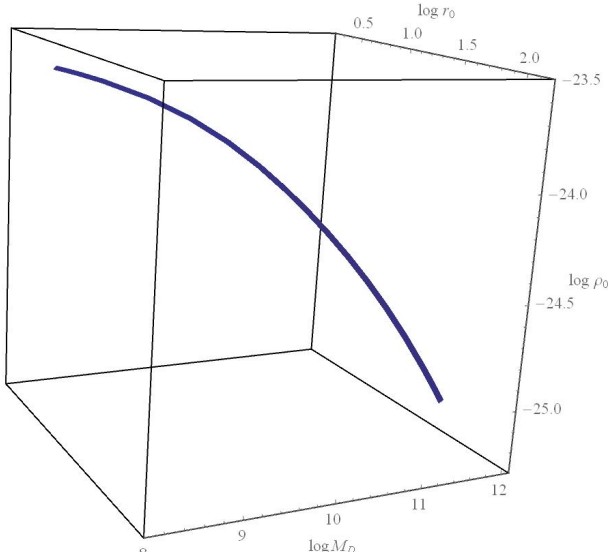

**Figure 4.** The relationship linking the DM and LM structural parameters $\rho_0, r_0, M_D$ (see [14]). Log-units: $M_\odot$, kpc, g/cm$^3$.

## 4. Unexpected Relationships

Let us first remark that the properties of the internal structure of the disk galaxies, at the basis of this work, have been discovered and independently confirmed in a series of works since 1991, to which we direct the reader for further information.[5] In the present work, we adopt them as the motivation for originally proposing a paradigm shift in how we shall investigate the dark matter mystery.

### 4.1. Central Halo Surface Density

The quantity:

$$\Sigma_0 \equiv \rho_0 r_0$$

i.e., the central surface density of the DM halo, is found constant in objects of any magnitude and disk morphology (see Figure 5 and [5,22,23,34,37–39]):

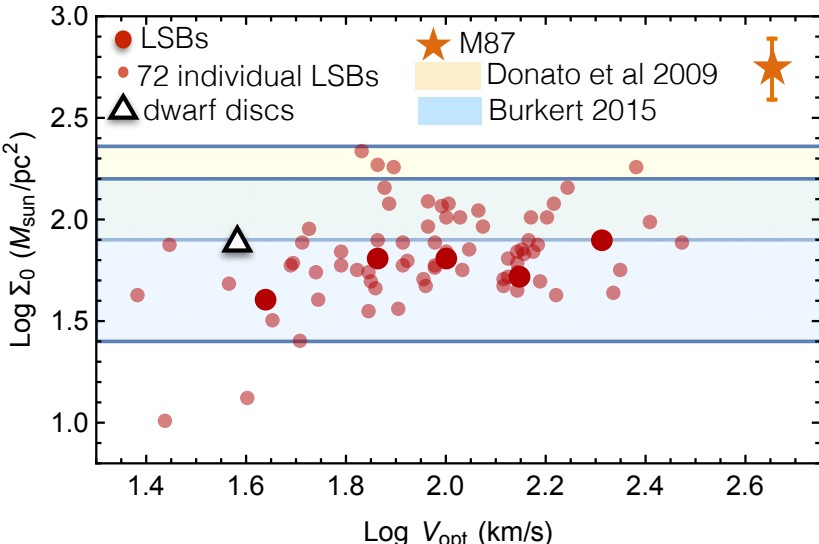

**Figure 5.** The dark halo central surface density $\Sigma_0$ as a function of the reference velocity $V_{opt}$ in disk systems and in the giant elliptical M87 (from [19]).

$$\text{Log} \, \frac{\Sigma_0}{M_\odot pc^{-2}} = 2.2 \pm 0.25 \tag{10}$$

this means that $\rho_0$, the value of the DM halo density at the center of galaxy, is inversely proportional to the size $r_0$ of the region in which the density is about constant. This seems to imply that the dark particle possesses some form of self-interaction of unspecified nature.

### 4.2. DM Core Radii vs. Disk Length Scales

Amazingly, since the pioneering work of [40][6] the core radius $r_0$ is found to tightly correlate with the stellar disc scale length $R_D$ [18–20,23,41],

$$\text{Log } r_0 = (1.38 \pm 0.15) \text{ Log } R_D + 0.47 \pm 0.03 \tag{11}$$

see Figure 6.

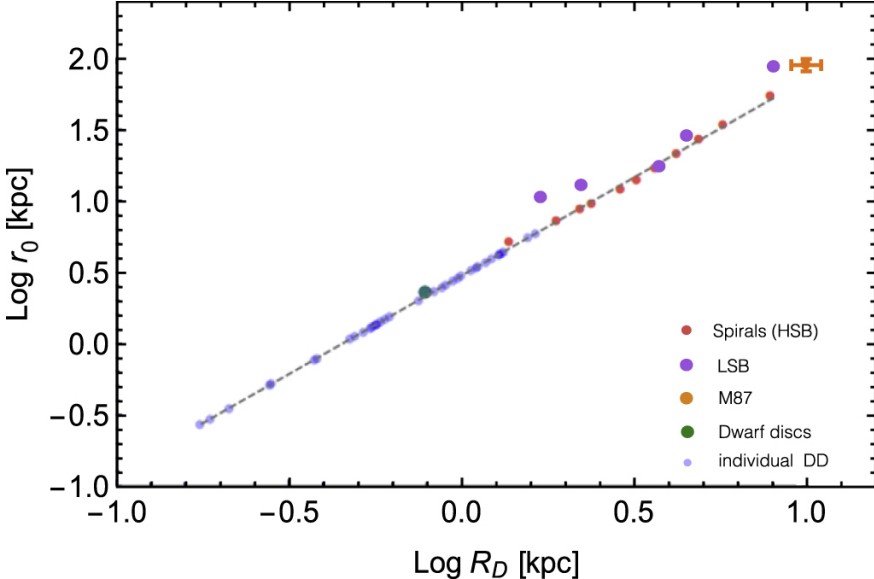

**Figure 6.** DM halo core radius $r_0$ (see Equation (8)) vs. the stellar disk length scale $R_D$ in Spirals, Dwarf Disks, Low Surface Brightness and the giant cD galaxy M87 (from [23]).

This relationship, initially found in Spirals, has also emerged in LSBs, Dwarf Irregulars and in the giant elliptical M 87 (see Figure 6). Overall, it extends in objects whose luminosities span over five orders of magnitudes. Then, the size of the region in which the DM density does not change (much) with radius, is found to be related with the size of the stellar disk $R_D$. It is very difficult to understand such tight correlation between very different quantities without postulating that dark and luminous matter are able to interact more directly than via the gravitational force.

### 4.3. Stellar Disks vs. DM Halos Compactness

Similar mysterious entanglement emerges also from the evidence that, in galaxies with the *same* stellar disk mass, the more compact is the stellar disk, i.e., the larger is the value of $C_\star = M_D/R_D^2$, the more compact results the 2-D DM density projected on the core region, i.e., the larger is the value of $C_{DM} = M_h(r_0)/r_0^2$ (see Figure 7, details in [19,20]. More globally, the stellar and the DM surface density, once they are estimated inside $r_0$, are found to be proportional [42]). Again, the dark and luminous worlds seem to have communicated in an unknown language.

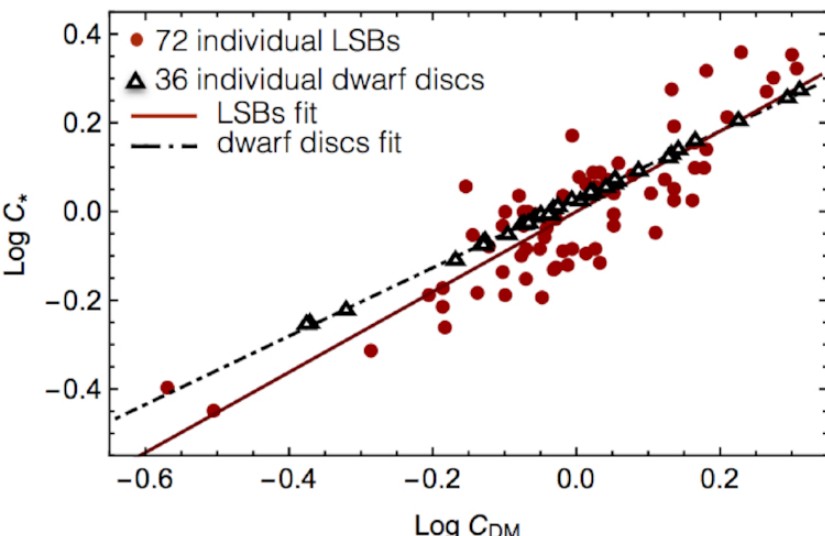

**Figure 7.** The compactness of the stellar disks vs. the compactness of DM halos, in units of their average values, in two different samples of galaxies, (see [19]) for details.

*4.4. Total vs. Baryonic Radial Accelerations*

Even without assuming a-priori the presence of a dark halo in galaxies, the dark halo emerges and shows a mysterious entanglement with the baryonic component. One can for instance consider $V^2(y)/y \equiv g$, the radial acceleration of a point mass in rotational equilibrium at a distance $y$ from the center of a disk galaxy, and $V_b^2(y)/y \equiv g_b$, its baryonic (stellar) component. In spiral galaxies we find $g(y) > g_b(y)$, that calls for a dark component, but also $g = g(g_b)$: the two accelerations are thus related quite tightly [43]. Including in the analysis also dwarf Irregulars and Low Surface Brightness galaxies, the above relation gains an other parameter, the radius $y \equiv R/R_D$.[7] The points with coordinates $(g, g_b, y)$ are found to be very well reproduced by a smooth surface $log\, g = \tilde{g}(log\, g_b, y)$ (see Figure 8). More specifically, in all galaxies and at all radii, the individual points lay distant from the average relationship by not more than 0.04 dex [44]. In a pure collisionless scenario, the origin of this thin surface, built by a fine tuning of dark and luminous quantities, is extremely difficult to understand.

*4.5. The Crucial Role of $r_0$*

The relationships above indicate the quantity $r_0$ as the radius of the region inside which the DM–LM interaction takes or has taken place. Let us show further direct support for such identification. In the self-annihilating DM scenario the number of interactions per unit time has a dependence on the DM halo density given by: $K_{SA}(R) = \rho_{DM}^2(R)$; in analogy, in the scenario featuring DM-baryons interactions (absorption and/or scattering), we focus on the quantity $K_C(R) \equiv \rho_{DM}(R)\rho_\star(R)$ which has no physical role in a collisionless DM particle scenario. From the above URC mass model we get a striking relation when evaluating $K_C$ at $r_0$:

$$K_C(r_0) \simeq const = 10^{-47.5 \pm 0.3}\, \mathrm{g}^2\, \mathrm{cm}^{-6} \qquad (12)$$

Impressively, we see in Figure 9 that the kernel $K_C(R)$, at any given physical radius $R$, varies largely (i) among galaxies of different mass, and, (ii) in each galaxy, at different radii.

Instead, at $R \simeq r_0$ and only there, this quantity takes the same value in all galaxies. In the scenario of interacting dark matter, this clearly suggests the radius $r_0$ as the edge of the region inside which interactions between dark matter particles and a Standard Model particles have taken place so far, flattening the original halo cusp.

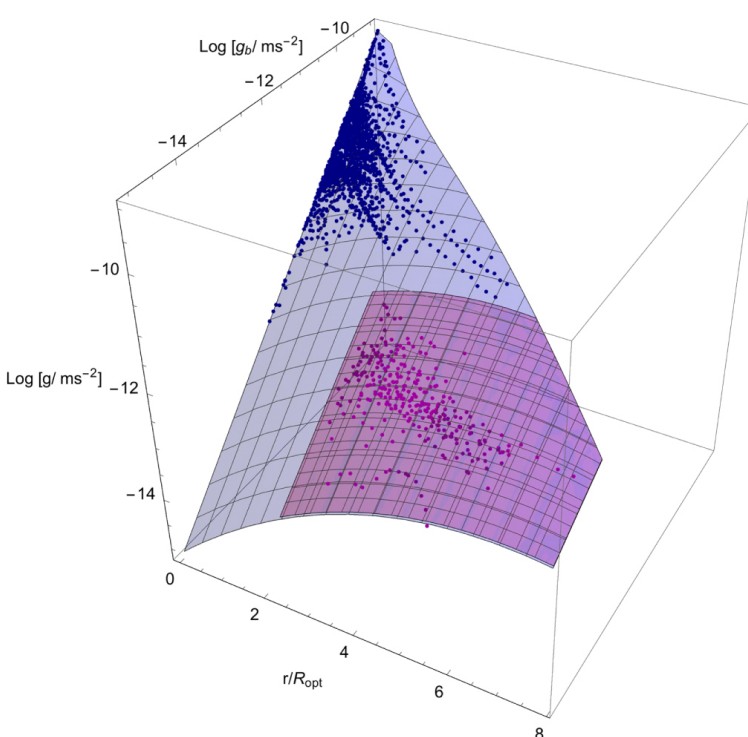

**Figure 8.** The amazing relationship in dwarf (dark blue) and LSB (blue) galaxies among (1) the total and (2) the baryonic acceleration (both evaluated at the radius $y \equiv R/R_D$) and (3) $y$. Points represent the values derived from the RCs (see details in [44]).

Let us notice that, at small scales, there are further observational evidences that cannot be framed by a scenario featuring a *collisionless* and *simple* dark particle [45]. Furthermore, in the ΛCDM scenario, at large scales and at high $z$, tensions of different types exist (e.g., [11]).

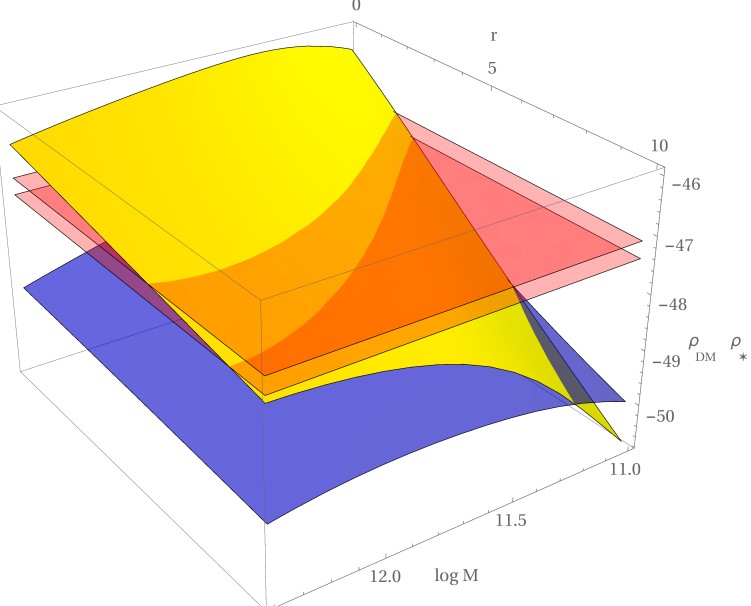

**Figure 9.** The kernel $\rho_{DM}(R)\rho_{LM}(R)$ as function of radius and halo virial mass (yellow). In all objects the value of the latter, at the boundary of the constant density region $r_0$, lies inside the two red planes. Also shown $\rho_{DM}(R)^2$ (blue) relative to the dark particle annihilation. Units: log $M_\odot$, kpc, log (g$^2$ cm$^{-6}$) (from [13]).

*4.6. Discussion*

Dark Matter particles have been originally envisioned with the crucial characteristic of interacting with the rest of the Universe essentially only by Gravity. However, once we set in such a framework, we realize that the properties of the mass distribution in galaxies do not make much sense for explaining the observed properties. An other interaction has to be considered. Remarkably, this interaction causes no effect on the structure of the galaxy dark halos on the time scale of their free fall, the one governing the WIMP particles. It acts within a timescale as long as the age of the Universe, by slowly modifying the dark halo density distribution.

It is worth, before proceeding, to discuss the possibility of a coexistence between the ΛCDM scenario and the above observational evidences. The best chance for this to work seems to be an astrophysical effect leading to the formation of the DM halo cores, via a global feedback created by explosions of galactic supernovae (e.g., [46]). We stress that, for this process and for any other with the same aim, the most serious trouble is not the efficiency in the core-forming process, but the ability to build up from scratch the above very complex and fine-tuned observational scenario.

In addition, there are also specific issues affecting the core-forming role of the baryonic feedback. According to the latter, in objects with the same stellar mass, one should find that the *more compact* is the stellar distribution (and consequently the *more efficient* is the process of removing the DM particles from the original cusped halo by a greater number of supernovae explosions) and the *less compact* the DM halo should be. This is in strong disagreement with present day observations (see Figure 7). Similarly, LSB galaxies, where the number of supernovae per unit area had been *much smaller* than normal spirals, are instead found to possess a DM core of size *larger* than that of the Spirals of the same mass (see Figure 6).

Finally, we detect Dark Matter cores also in dwarf spirals [20], in giant LSBs [19] and ellipticals [23], i.e., in situations where the SN explosions have been too few or where the gravitational potential is too strong to allow for a baryonic feedback flattening the primordial cusps (see e.g., [47]).

As a result, the idea of bringing observations in line with the standard DM scenario of collisionless particles via astrophysical processes, seems to have essential problems.

## 5. A New Paradigm

The impact of the above observational scenario goes beyond the evidence of its tension with the Λ*CDM* WIMP theoretical scenario. In fact, the disagreement between the two scenarios is so strong and so deep that we are led to think that it can rule out the Apollonian paradigm itself (from which the Λ*CDM* scenario has emerged). The same defining criteria (1)–(5) of the paradigm appear unable to account for the above observational evidence. Thus, the spectacular DM-LM entanglement found in galaxies, allied with the fact that the WIMP particle has escaped detection, becomes a strong motivation for demanding a shift of the Paradigm that we shall follow to approach the dark matter Phenomenon and determine the nature of the dark particle.

Reflecting upon the failure of the current paradigm, we realize that it originates from the fact that it forces any scenario created to explain the DMP under its ruling, to have inbuilt a direct positive correlation between truth and beauty. On the contrary, the observational properties of the dark and luminous matter in galaxies seem to favor scenarios which may appear "ugly". Indeed, the found observational relationships and the galaxy properties seem to indicate that the (proper) theoretical scenario for the DMP may have a large number of free parameters, a limited predictive power, no obvious connection with known Physics, or *expected* new Physics, including the currently open issues in Fundamental Theoretical Physics. Then, the true scenario could likely be at odds with the entire Apollonian paradigm.

In other words, we need a new Paradigm that opens the door to "ugliness", thus allowing scenarios for the DMP that are forbidden by the current Apollonian Paradigm.

Many philosophers have expressed their interest in situations like this; most notably, F. Nietzsche [48] has been obsessed by the concepts of beauty and ugliness in relation to those of truth and falsity, so we name after him the proposed new Paradigm.

Thus, we claim that, in order to formulate the correct scenario for the DMP, we need to abandon both the $\Lambda CDM$ *scenario* and its generating Apollonian *paradigm* and to adopt the newly proposed Nietzschean *paradigm*.

This new paradigm: (i) *values* and (ii) *protects* from negative biases any theoretical scenario for the DMP that emerges from observations even if it appears exotic, complex or full of mysterious entanglements. Then, it directs our investigations according to the following loop: reverse-engineering the available observations leads us to a DM scenario that gets tested by a *new* set of especially selected observations. Reverse engineering the old and the new observation improves then the scenario. The paradigm affirms that that after some iteration, the actual scenario for the DMP will emerge and reveal, at the same time, the nature of the dark particle.

Before proceeding, let us stress that the proposed paradigm shift is not an straightforward and painless step. In fact, the old paradigm has created the $\Lambda$CDM scenario which has a number of clear advantages:

- The underlying Physics is rather simple and at the same time is connected with new Physics in the fields of Cosmology and Elementary Particles.
- When it is adopted, the  initial conditions and the theoretical framework at the basis of any new investigation are well-established.
- It has a clear agenda for the investigation of the dark matter mystery, already in use in the scientific community and fostering a global spirit of research.
- It connects "state of the art" computer simulations, observations and experiments.

Therefore, to abandon the Apollonian paradigm and, in turn, the generated $\Lambda$CDM scenario, has important consequences in the investigation of the DM phenomenon. In fact, we do not have yet a scenario ready to take the role that the $\Lambda$CDM scenario has played so far. More specifically: from the available observational evidence collected so far, we can definitely argue that the true scenario behind the DMP will result much more complicated, complex in its background physics and less able to take advantage of computer simulations than the current $\Lambda$CDM scenario. Moreover, very likely, no other future scenario will profit of the united effort of the large majority of cosmologists, as it happened for the $\Lambda$CDM one. Given this, it is not possible to sneak away from the $\Lambda$CDM scenario to some other scenario without performing a deep rethinking that involves also the generating Paradigm.

Summarizing, we propose a new Paradigm according to which the search for the true DMP scenario can violate or/and go beyond the five points in Section 2, but, on the other hand, must reverse-engineer the available observational and experimental data.

## 6. Uncharted Territories?

We complete the goal of calling for a DM paradigm switch by showing that, effectively, the new paradigm outlined above is able to provide us with promising scenarios. Within this, the first relevant observation to be made is that all the correlations emerging between luminous and dark parameters appear to be essentially a manifestation of some (new) physics taking place at galactic scales as it is clear in the outstanding issue of the formation of galactic cores. In the search for the true DM scenario it is intriguing that, within the new Nietzschean paradigm, we are allowed to speculate that the detected dark-luminous relationships are just the consequence of a non-standard interaction between DM and baryons and, above all, to proceed by neglecting the constraints (1)–(5), whose obedience has limited so far the birth and growth of scenarios alternative to the DM. More specifically, we can start to consider of the following scenarios:

- Scenario for which the baryon-only physics, in various forms of feedback, by means of (a likely complex) energy release is able to modify the DM distribution in galaxies. If it includes collisionless dark particles it has clear difficulties in accounting for the DM-DM and DM-baryons relations described above, however, these difficulties

are likely to disappear if we postulate *also* the presence of a *proper* SM particle-DM particle interaction.

- Scenario in which a new direct Baryon-DM interaction is responsible for the core formation. A simple estimate assuming a total dark matter core transmutation, leads to a quite high value for the relative cross section, $\sim 10^{-24,25} \text{cm}^2 \sim 0.1$–1 barn. This might be considered not realistic, but let us note that if we consider the *dynamical* evolution in the dark halo particles, this may help to reach an adequate transfer of energy from the LM to the DM component also with a much smaller interaction cross section.
- Scenario featuring a DM-DM interaction whose existence and value of its cross section derive from the detection (in galaxies) of a roughly constant value for the DM surface density inside the core radius $r_0$.of $\Sigma_0 \simeq 100 \, \text{M}_\odot / \text{pc}^2$ for the DM surface density inside the core radius $r_0$. This leads to a quite large cross section of $\sigma/m = 1 \, \text{cm}^2/\text{g} = 1 \, \text{barn/GeV}$, but again, a proper treatment of the evolution of the dark matter halos at short scales could reveal that also smaller cross sections are effective in core formation, turning on some gravitational energy transfer between the dark and the luminous components.
- Scenario in which the core-forming dark-luminous interactions occur (in a time scale of 10 Gyr) inside or at the surface of bound objects like individual or binary stars, white dwarfs, BHs of any mass and their accretion disks, planets and their atmospheres and supernovae expanding shells, i.e., in realistic places that, however, have not been theoretically and observationally explored so far.
- The scenario featuring a WIMP particle + baryonic feedback can likely come back into the play if inserted in a modified gravity frame.

## 7. Conclusions

Here, we have motivated our proposal according to which, in the investigation of the complex and entangled world of the phenomenon of the Dark matter in galaxies, we take a new and tailored approach.

In detail, we advocate for a paradigm according to which, after abandoning the failing ΛCDM scenario, we must be poised to search for scenarios without requiring that: (a) they naturally come from (known) "first principles" (b) they obey to the Occam razor idea (c) they have the bonus to lead us towards the solution of presently open big issues of fundamental Physics. On the other side, the proper search shall: (i) give precedence to observations and the experiment results wherever they may lead (ii) consider the possibility that the Physics behind the Dark Matter phenomenon be disconnected from the Physics we know and and does not comply with the usual canons of beauty. Finally, as regard of the impact of this work in the scientific community, it is irrelevant whether such a search is undertaken to follow the proposed paradigm shift or as consequence of a more agnostic approach regarding any paradigm for the DMP.

**Author Contributions:** All authors contributed equally to this investigation. All authors have read and agreed to the published version of the manuscript.

**Funding:** This research received no external funding.

**Conflicts of Interest:** The authors declare no conflict of interest.

## Notes

1. In this work, for simplicity, we use the present tense also in reporting published and well known results.
2. $\star$ and $h$ refer to the disk and the halo component. Let us define: HI = neutral hydrogen. DM = Dark Matter. DMP = Dark Matter Phenomenon. RC = Rotation Curve. SM = Standard Model of elementary particles. LHC = Large Hadron Collider (CERN). ΛCDM = Lambda CDM cosmological model. WIMP = Weakly Interacting Dark Matter Particle. Apollonian (philosophy) = ideas from the famous Greek school of philosophy. Nietzschean (philosophy) = (some) ideas from the German philosopher.
3. For simplicity, we neglect here the small contribution of the HI gaseous disk.
4. The DMP is the ensemble of all the available cosmological and astrophysical observations which result not existing in an Universe made of only SM particles.

5  References in this work and in the review [5].

6  See their Equation (9a) in combination with the Equation (8) above.

7  Notice that the new parameter is not the expected physical galactocentric radius $R$, but this quantity *normalized* to the length-scale of the galaxy stellar disk $\propto R_D$.

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
