# Peer review of "The Quest for the Nature of the Dark Matter: The Need of a New Paradigm"

_2674-0346, doi:10.3390/astronomy2020007_

Round 1

Reviewer 1 Report

This paper suggests we need a new paradigm for dark matter, but does not say what that new paradigm should be.  It reads more like a review article than an original research paper.  I do not recommend publication.

Author Response

We reviewed the paper and  better expressed/explained  (especially in section 5) our new paradigm. The work  is original in that it discusses well known observational and experimental facts and then proposes a new way to del with the dark matter phenomenon  that emerges from them. The review approach of the first sections is obliged since it consists of a crucial display for our new result . 

Reviewer 2 Report

The submission is essentially a review describing some of the problems with the paradigm of dark matter. There are some novel aspects of the presentation (I appreciated in particular the introduction, section 4 and section 6), however there are some problematic aspects that should be amended before it can be published.

The first is that the language is at time hyperbolic and/or unclear. Section 5 in particular contains several false statements: "It is certain that the impact of these observational evidences goes beyond the falsification of the ΛCDM WIMP scenario ..." etc and the rest has an unclear meaning. I would suggest that that section simply be deleted, or very heavily rewritten.

The second problem is that there is no attribution to many of the plots. I believe that they are probably from the authors' papers in most cases, but the references in the figures are (mostly) not given.

Finally, there is some significant overlap in subject with the extensive review "From Galactic Bars to the Hubble Tension: Weighing Up the Astrophysical Evidence for Milgromian Gravity" by Banik and Zhao, published in the MDPI journal symmetry. At least this work deserves to be cited, but it should also be properly referred to.

Author Response

We thank the referee for her/his helpful comments.

In the light of the first point  made by the referee. we have, indeed, rewritten great part the section 5 and eliminated some  hyperbolic and not completely useful  statements  to better explain our point of view. 

Then, we have improved all the captions in order to make easier for the reader retrieve the original work.

Finally, we  have also properly cited the suggested work.

Reviewer 3 Report

This paper gives a concise analysis of the dark matter paradigm and suggests a paradigm shift. Basically, 5 different unexplained observational correlations are discussed, with the conclusion that the DM must interact with itself and with the SM particles. The authors conclude their work by stating the need for a new paradigm (a Nietzsche approach in place of an Apollonian approach). This paper can be useful for research on DM. I have few concerns before recommending it for publication:

1) The use of the symbol "\nabla" for two different objects in closely-related equations just above Eq. (4) can confuse especially the beginners. It could be enlightening to explain Eq. (4) and explain also symbols like "\nabla(2)". Does it mean "y=2"?

2) In discussing unexplained relationships in p. 12, the authors conclude that the DM halo distribution must have changed on the Hubble scale not on the scale of free-fall time around galaxies. This means that DM-LM couplings must be exceedingly small.  The authors do not attempt to explain such a coupling scheme. They mainly state properties pointing uncharted territories in Sec. 6. This is an important conclusion and needs a little bit explanation or suggestions from the authors' side. In fact, this finding reminds me of the paper:

D.~Demir, %``Naturally-Coupled Dark Sectors,'' Galaxies \textbf{9} (2021) no.2, 33 doi:10.3390/galaxies9020033 [arXiv:2105.04277 [hep-ph]].

in which the author was studying dark sectors (comprising the DM) that do not destabilize the electroweak scale, and was concluding that the DM-LM coupling should go like M_higgs^2/M_DM^2 when M_DM > M_higgs and like M_DM^2/M_higgs^2 when M_DM < M_higss. These   suppressed couplings might comply with authors' findings. 

3) The DM halo core radius is denoted by r_c in Fig. 6. Maybe, authors can use a uniform notation r_c everywhere (in place of r_0).

4) The caption for Fig. 8 could be improved by explaining the data points as well.  

5) The authors' claim that dynamical evolution in the halo can enhance feeble interactions (to the level of static interactions with significant couplings) needs be detailed. Namely, the second and third uncharted territories in p. 14 need be detailed further. 

6) The authors may consider performing a grammar check of their paper.

Author Response

We thank the referee for the very useful comments.

1) we have denoted the Laplacian as \Delta.

2) It is not generally true that a slow timescale of the DM halos evolution implies an exceedingly small coupling with visible matter. Indeed, this statement will need to be clarified in a quantitative way, in that the interaction can depend on various environmental factors, like le  velocities of the two components, the relative densities, etc. This kind of detailed analysis will clearly be the subject of all the future activity in pinpointing the right scenario. For this reason we also refrain, at this stage, from dwelling into any particular particle physics theoretical arguments, in accordance with our paradigm that gives precedence to phenomenology.  It will be nevertheless interesting in the future to consider also directions like the one you pointed us.  Finally we have improved the paper to make clearer the above .

3)-4) We have updated figure 6 and corrected the captions.

5) We agree  that discussion was too concise, and now  we have clarified the points. Our estimate of the DM-SM cross section was under the scenario of an interaction that totally converts the DM into baryons. Clearly this is not the only option: as an example already proposed by various authors, a much smaller cross section for interactions which heat the DM distribution, can lead to an equivalent expulsion of the DM cusp. Similar discussions apply to the DM-DM interactions, studied under the name of Self Interacting Dark Matter (SIDM). 

6) we have checked and improved the English syntax and grammar.

Round 2

Reviewer 2 Report

The authors have addressed the concerns in my previous report and I now recommend this review paper for publication.

Reviewer 3 Report

The authors made necessary improvements in the manuscript. I recommend publication of this paper in the present form.